# PeerJ

# The awareness of novelty for strangely familiar words: a laboratory analogue of the déjà vu experience

Josephine A. Urquhart and Akira R. O'Connor

School of Psychology & Neuroscience, University of St Andrews, UK

## ABSTRACT

Déjà vu is a nebulous memory experience defined by a clash between evaluations of familiarity and novelty for the same stimulus. We sought to generate it in the laboratory by pairing a DRM recognition task, which generates erroneous familiarity for critical words, with a monitoring task by which participants realise that some of these erroneously familiar words are in fact novel. We tested 30 participants in an experiment in which we varied both participant awareness of stimulus novelty and erroneous familiarity strength. We found that déjà vu reports were most frequent for high novelty critical words ($\sim$25%), with low novelty critical words yielding only baseline levels of déjà vu report frequency ($\sim$10%). There was no significant variation in déjà vu report frequency according to familiarity strength. Discursive accounts of the experimentally-generated déjà vu experience suggest that aspects of the naturalistic déjà vu experience were captured by this analogue, but that the analogue was also limited in its focus and prone to influence by demand characteristics. We discuss theoretical and methodological considerations relevant to further development of this procedure and propose that verifiable novelty is an important component of both naturalistic and experimental analogues of déjà vu.

Corresponding author
Akira R. O'Connor,
aro2@st-andrews.ac.uk

Déjà vu is defined as a "clash between two simultaneous and opposing mental evaluations: an objective assessment of unfamiliarity with a subjective evaluation of familiarity" (p. 2, *Brown, 2004*). The sensation has captured public interest, e.g., its discussion in *Heller*'s (*1961*) 'Catch 22' and use as a plot device in 'The Matrix' (*Silver, Wachowski & Wachowski, 1999*), but its scientific investigation remains sparse, perhaps because the sensation is fleeting and occurs unpredictably. Despite its volatility, the experience is by no means uncommon—surveys usually find lifetime incidences in excess of 65%, with young adults likely to report multiple yearly experiences (*Brown, 2003*). As such, déjà vu presents a window into the healthy memory decision-making process through study of perturbations to the signals it must adjudicate between.

Much of the research that has examined déjà vu has used the clinical case study approach (e.g., *Moulin et al., 2005*; *Bancaud et al., 1994*) or employed retrospective report to explore individual differences (*Martin et al., 2012*; *O'Connor & Moulin, 2013*). Clinical studies allow the study of déjà vu-like experiences in samples for which they

form part of a regularly occurring constellation of symptoms associated with a primary disorder such as dementia or epilepsy. However, the correspondence between clinical and nonclinical manifestations of déjà vu remains unclear, with the potential that they may be mechanistically and subjectively different (e.g., clinical déjà vu associated with dementia, termed déjà vécu, has behavioural consequences—unlike déjà vu in the healthy population, patients with déjà vécu modify their behaviour to avoid the sensation of déjà vu; *Moulin et al., 2005*). Retrospective reports, whilst affording study of déjà vu in the healthy population, are often recovered weeks or months after the experiences occurred, leaving them open to contamination by bias and reconstruction (*Chapman & Mensh, 1951*). Consequently, there has been a recent drive towards developing laboratory-based procedures which reliably generate déjà vu in healthy volunteers. Such laboratory analogues provide the opportunity for the 'here-and-now' study of nonclinical déjà vu experiences and could yield insights into memory decision-making akin to those offered into word-finding by the experimental generation of the tip-of-the-tongue sensation (e.g., *Widner, Smith & Graziano, 1996*; *Schwartz, 2001*).

Attempts to find a laboratory analogue of déjà vu have primarily focused on generating sensations of subjective familiarity. For example, *Brown & Marsh (2009)*; building on *Jacoby & Whitehouse (1989)*, using subliminal presentation of symbols, and *Cleary, Ryals & Nomi (2009)*, using configural similarity for visual scenes, both generated familiarity in the absence of awareness of its source. The frequency of déjà vu reports stemming from these procedures was high, and the willingness of participants to categorise the experimentally-generated experience as déjà vu likely reflects an overlapping experiential feature, familiarity without recollection. Nevertheless it should be noted that experiences more closely analogous to this experimentally-generated sensation, at least in their causal mechanism, occur frequently without being labelled as déjà vu e.g., *Mandler*'s (*1980*) example of the 'butcher on the bus' (formalised in the laboratory as recognition without identification, *Cleary, 2006*). In the 'butcher on the bus' experience, an individual becomes aware that they recognise someone, but cannot recollect who the person is because the person (the butcher) is being encountered in a different context to that which they were previously encountered (on the bus as opposed to in the supermarket). In the *Brown & Marsh (2009)* and *Cleary, Ryals & Nomi (2009)* experiments, participants may have misidentified the sensation of recognition without identification as déjà vu because both experiences represent unusual memory sensations where retrieval feels incomplete. Our rationale for continued work towards a laboratory analogue is that a compelling elicitation of déjà vu should attempt to generate all of the components of the experience. In order to do this, we refer again to the definition presented in the first paragraph which incorporates subjective familiarity but also a concurrent awareness of objective unfamiliarity.

The key omission in the déjà vu generation procedures described above is the provision of information allowing the participant to make an evaluation of unfamiliarity or novelty to clash with the experimentally-generated familiarity. In these procedures, there was no objective standard by which participants could verify that the stimuli provoking familiarity had in fact not previously been encountered. With a view to generating a more complete

laboratory analogue of naturalistic déjà vu, we developed a procedure during which some stimuli elicit both subjective familiarity and an awareness of novelty. This procedure builds on the DRM recognition task (*Deese, 1959*; *Roediger & McDermott, 1995*) in which participants study a series of words (e.g., rest, bed and blanket) which are all semantically linked to an unpresented word (sleep). This unpresented word, referred to as the critical lure, typically yields illusory recognition when it is presented at test—the critical lure generates a sensation of subjective familiarity. Our procedure pairs the DRM task with an additional task in which participants monitor studied stimuli for a feature present only in the critical lure (e.g., the starting letters 'sle'). When participants become aware of the absence of that feature from the study list words, they also become aware that the critical lure must in fact be novel (see Fig. 1). Thus in critical lures, subjective familiarity clashes with an objective awareness of novelty, satisfying the definition of déjà vu.

Using this new procedure, we independently varied objective novelty and subjective familiarity, hypothesising that déjà vu reports would be greatest within lists for which the greatest clash between familiarity and novelty was contrived. Assessing déjà vu occurrence on a trial-by-trial basis allowed us to identify the specific word triggers of déjà vu. We hypothesised that déjà vu triggers would be most numerous amongst the words for which maximal familiarity/novelty conflict was generated, critical lures. Finally, we supplemented our categorical déjà vu assessments with discursive responses which we acquired pre- and post-experimentally. We asked participants to write about (i) a previous typical naturalistic experience of déjà vu and (ii) the experimentally-generated experience of déjà vu according to the same criteria. We used these responses to better understand the similarities and differences between our experimentally-generated déjà vu experience and naturalistic déjà vu experiences.

## METHOD

### Participants

Thirty English-speaking participants (20 female, 10 male; mean age = 24.1 years, SD = 6.5 years) were tested and reimbursed at a rate of £5/h for their time. Written consent was obtained from all participants. The protocol was approved by the University Teaching and Research Ethics Committee at the University of St Andrews (approval number PS10697).

### Stimuli and materials

Over the course of the experiment, each participant was presented with 24 DRM word lists based on *Stadler, Rodiger & McDermott*'s norms (*1999*). The 24 lists comprised the 12 which yielded the highest false alarm rates for critical lures and the 12 useable lists which yielded the lowest false alarm rates. For each study list, 12 words were randomly selected from the 15 words published per list in Stadler et al. For each test list, 3 old words (targets; selected from the previously studied 12 words) were presented alongside 2 semantically unrelated new words (unrelated lures), 2 semantically related new words (related lures; these were randomly selected from the 3 words excluded from study presentation) and

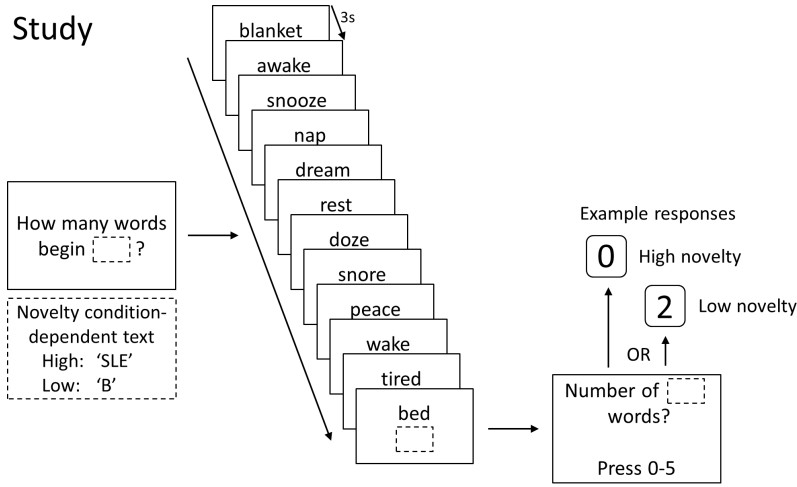

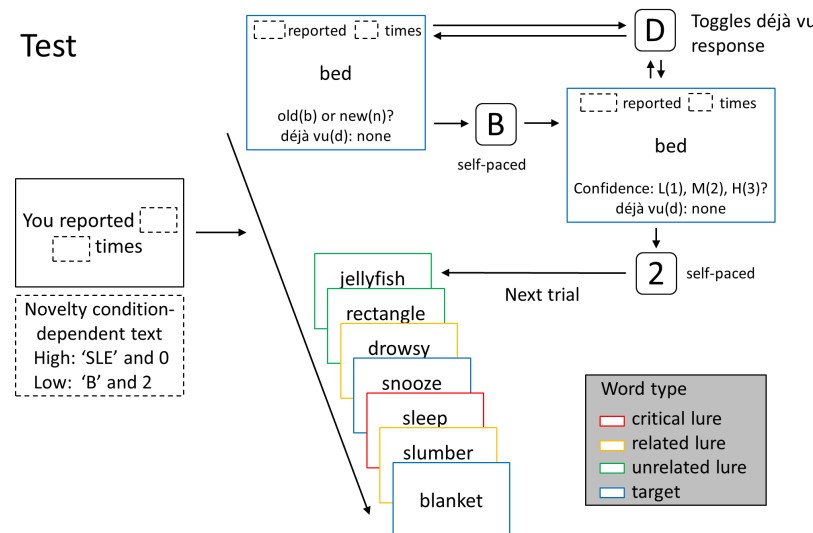

**Figure 1 Schematic of study and test lists.** A possible study and test list for a high familiarity DRM list with critical lure 'sleep'. At the start of the study list, participants were presented with a question reminding them to monitor the study list for words beginning with a character string. The character string remained onscreen throughout the study list. In this case, the high novelty condition string was 'SLE' and the low novelty condition string was 'B'. Participants were then presented with 12 words semantically related to the unpresented critical lure, each word presented at 3 s intervals. At the end of the study list, participants indicated the number of words presented which began with the character string. In high novelty conditions, the correct answer was always '0', in low novelty conditions the correct answer always greater than 0. Six study lists were presented in each study phase before the test phase was initiated. At the start of each test list, participants were reminded of the previously monitored string and the number of words they reported at study beginning with this string. The reminder remained onscreen throughout the eight-word test list. For each test trial, participants first indicated whether the word had been presented at study (old) or not (new) and then indicated their confidence in this decision. Throughout both of these self-paced decisions, participants could toggle their déjà vu response from (continued on next page...)

**Figure 1 (...continued)**
'none' through 'low', 'medium' and 'high'. Once participants made a confidence decision the next test trial was initiated. In the schematic, different word conditions are shown bounded by different coloured boxes and the reminder information is omitted from test words 2–8 for the sake of clarity. In the experiment there was no differentiation of stimulus type visible to participants. The six test lists corresponding to the six study lists from the preceding study phase were presented in the test phase. Over the entire experiment, there were four study-test blocks.

the single critical lure (see Appendix S1 for four example study-test lists). All word lists were presented on PCs running MATLAB (The MathWorks Inc., Natick, MA, 2000) and Psychophysics Toolbox (*Brainard, 1997*).

Two paper questionnaires were administered: a pre-experimental questionnaire; and a post-experimental questionnaire. The pre-experimental questionnaire was completed before the first study-test block and assessed participants' previous déjà vu experience (yes/no response) and frequency (<1, 1–4 or >4, times a year). There followed an explanation of déjà vu based on *Brown*'s (*2004*) definition, "a feeling of familiarity coupled with the knowledge that this familiarity is incorrect", and an open-ended question requested that participants write a short passage summing up previous déjà vu experiences according to the following instructions: "Please provide a short account of a 'typical' déjà vu experience you have had. Try to include some detail concerning the following points: Emotional intensity of a typical déjà vu experience; Duration of a typical déjà vu experience; How a déjà vu experience might typically make you feel about the reliability of your memory". The post-experimental questionnaire, completed at the end of the experiment, confirmed participants' experiences of déjà vu during the experiment (yes/no response) and was again followed by an open-ended question requesting a summary of the experimentally-generated experience of déjà vu according to the following instructions: "Please provide a short account of your déjà vu experience(s) during the experiment. Try to include some detail concerning the following points: Emotional intensity of déjà vu experience(s); Duration of déjà vu experience(s); How the déjà vu experience(s) made you feel about the reliability of your memory".

## Design and procedure

We manipulated subjective familiarity by presenting participants with DRM lists yielding the highest or the lowest critical lure false alarm rates according to *Stadler, Rodiger & McDermott (1999)*. Lists yielding high false alarm rates were used in the high familiarity conditions. Lists yielding low false alarm rates were used in the low familiarity conditions. We manipulated objective novelty by varying the string that participants monitored study words for. For each study list, participants were presented with a new 1–3 character string, and indicated the number of words which began with this string once they had seen all 12 words. For the high novelty lists, no words in the study list and only the critical lure in the test list began with the character string. For the low novelty lists, at least one word in the study list (and not the critical lure) began with the character string. During each test list, the number of words indicated by participants as beginning with the string was re-presented to participants, highlighting the novelty of critical lures in only the high

novelty condition. Thus, there were two within-subjects list-level factors, novelty (high, low) and familiarity (high, low). These combined to produce four types of list: (1) high novelty-high familiarity; (2) high novelty-low familiarity; (3) low novelty-high familiarity; and (4) low novelty-low familiarity. There was also one within-subjects word-level factor with four levels: targets (previously studied words), unrelated lures (previously unstudied words which were semantically unrelated to the studied words), related lures (previously unstudied words but which were drawn from the same DRM list), and critical lures (previously unstudied words to which all the studied words were semantically related). All four word conditions were presented within each test list.

Over the course of the experiment, six study-test list pairs of each list type (high novelty-high familiarity etc.) were presented (24 study-test list pairs in total). To allow participants to rest periodically, the experiment was split into four study-test blocks. The list composition of each block was randomly assigned such that participants would not inevitably encounter each list type in each study-test block. Each study-test block comprised six consecutive study lists, followed by six consecutive test lists. For each participant, corresponding study and test lists were presented in the same order (i.e., study1, study2, study3, study4, study5, study6, test1, test2, test3, test4, test5, test6).

Figure 1 shows a schematic of a study list and its corresponding test list. The to-be-monitored character string was presented in size 48pt font alongside word stimuli for the duration of each study list. Twelve words were serially presented in size 48pt font, in the centre of the screen, for 3 s each. At the end of the study list, participants were prompted to register the number of words beginning with the character string in that list (0–5 indicated using the keyboard).

Throughout each test list, the previously-monitored character string and the number indicated by the participant at the end of the study list were presented in size 48pt font, at the top of the screen. Eight words (three targets, five lures) were then serially presented in size 48pt font, in the centre of the screen. Test words were presented in a pseudorandom order modelled on the procedure used by Roediger & McDermott (1995). A target always occupied test position 1 and the critical lure always occupied test position 6, 7 or 8. (The three targets selected comprised the word from study position 1, one word selected at random from study position 2–6 and one from study position 7–12.) The remaining targets and lures were allocated at random to the unoccupied test positions. Below each word, the prompt, "old(b) or new(n)?", presented in size 48pt font, cued participants to indicate whether the word was previously presented at study or not. Once a response had been made, a new prompt, "Confidence: L(1), M(2), H(3)?", cued participants to indicate their confidence in the previous decision using. All test responses were self-paced and responses were made using the keyboard keys listed in parentheses.

In addition to the old/new and confidence judgments collected for each test stimulus, we also assessed the occurrence of a déjà vu experience and its intensity using an on-screen toggle system. This response system avoided unnecessarily asking participants about their déjà vu experiences repeatedly, leaving them free to report the experience only when it arose. A déjà vu status bar, located at the bottom of the screen had the default status "deja

vu(d): none". Participants reported the occurrence of déjà vu by pressing the 'd' key, which would cycle through the intensity options. On pressing it once, the status would change to "deja vu(d): low". Pressing it again would result in a status change to "deja vu(d): medium" and a third time, to "deja vu(d): high". If the 'd' key was pressed a fourth time, this would restart the cycle at "deja vu(d): none". Participants could indicate the occurrence of déjà vu at any point during each trial (i.e., during the self-paced windows for old/new and confidence responding). The status reverted to the default "deja vu(d): none" at the start of each new test word presentation. When participants had completed four study-test blocks they completed the post-experimental questionnaire. The entire procedure lasted no longer than one hour for each participant.

### *n*-gram analysis of déjà vu descriptions

*n*-grams are continuous sequences of *n* words found to occur within a passage of prose. *n*-grams with $n = 1$ are referred to as unigrams, and those with $n = 2$ as bigrams. Examples of unigrams from within this sentence are "sentence" and "from", whereas examples of bigrams from within this sentence are "of unigrams" and "are sentence". We identified differences in the strings of words used to describe naturalistic and experimentally-generated déjà vu by conducting a rudimentary *n*-gram analysis. The procedures reported here largely mirror those reported in *Selmeczy & Dobbins (2014)*. Prose passages from the pre-experimental and post-experimental questionnaires underwent identical preparation for *n*-gram analysis: spelling errors were corrected; contractions were completed (e.g., "don't" became "do not"); symbols with known meanings were written out in full (e.g., "/" became "or", "=" became "equals"); and "deja vu" replaced with "dejavu".

Separately for unigrams and bigrams, we counted the number of times each *n*-gram appeared in the pre- and post-experimental passages. For *n*-grams with total occurrences (*N*) across both passages of at least the median (unigrams: 5, bigrams: 4), we used *N*, the number of occurrences in the pre-experimental passage, and an assumed binomial-*p* parameter of .5 to calculate a binomial distribution *z* value and corresponding *p* value of each *n*-gram. As we were interested in *n*-grams which differentiated the two accounts, we set an uncorrected *p* threshold of .05 and tabulated these *n*-grams for examination.

## RESULTS

Whilst we do not present a comprehensive analysis of the accuracy data here—we present key analyses in the text and summarise accuracy and confidence for all conditions in Table 1—we show that the novelty and familiarity component manipulations of the modified DRM task resulted in the expected behavioural changes for recognition accuracy. We then examine the frequency of déjà vu reports, the intensity of déjà vu experiences and finish with a descriptive *n*-gram analysis of the responses to the open-ended questions.

### Accuracy

To establish that the DRM procedure was indeed generating erroneous familiarity for critical lures, we conducted a one-way repeated measures ANOVA on accuracy according to word condition (critical lure, related lure, unrelated lure, target), collapsed across list

**Table 1 Accuracy and confidence for old/new judgements and déjà vu likelihood according to condition.** Upper-case N indicates high novelty lists, lower-case n indicates low novelty lists. Upper-case F indicates high familiarity lists, lower-case f indicates low familiarity lists. Accuracy is expressed as the proportion of correct responses. Confidence is expressed as the mean confidence in recognition response accuracy, where response options 'low', 'medium' and 'high' were coded 1, 2 and 3 respectively. Déjà vu indicates déjà vu likelihood, expressed as the proportion of words eliciting a report of déjà vu (of any intensity). In all cells, means are shown, followed by 95% CIs in brackets.

| | List | | | | |
| | N/F | N/f | n/F | n/f | Overall |
|---|---|---|---|---|---|
| **Accuracy** | | | | | |
| Critical lure | .667 [.561, .773] | .761 [.662, .860] | .283 [.182, .384] | .544 [.446, .642] | .564 [.493, .635] |
| Related lure | .731 [.653, .808] | .811 [.739, .883] | .769 [.703, .836] | .839 [.776, .902] | .788 [.731, .844] |
| Unrelated lure | .925 [.882, .969] | .933 [.897, .969] | .947 [.914, .980] | .928 [.888, .968] | .933 [.903, .964] |
| Target | .759 [.707, .812] | .759 [.707, .812] | .754 [.694, .813] | .774 [.715, .833] | .762 [.715, .808] |
| Overall | .770 [.727, .814] | .816 [.772, .861] | .688 [.649, .728] | .771 [.732, .810] | .762 [.725, .798] |
| **Confidence** | | | | | |
| Critical lure | 2.34 [2.14, 2.54] | 2.50 [2.30, 2.70] | 1.93 [1.75, 2.12] | 1.97 [1.78, 2.17] | 2.19 [2.02, 2.35] |
| Related lure | 1.98 [1.79, 2.17] | 2.01 [1.82, 2.21] | 1.97 [1.79, 2.14] | 2.01 [1.83, 2.19] | 1.99 [1.82, 2.17] |
| Unrelated lure | 2.36 [2.16, 2.56] | 2.34 [2.14, 2.53] | 2.43 [2.24, 2.61] | 2.33 [2.16, 2.51] | 2.36 [2.18, 2.54] |
| Target | 2.25 [2.07, 2.42] | 2.39 [2.23, 2.54] | 2.30 [2.14, 2.46] | 2.42 [2.26, 2.57] | 2.34 [2.19, 2.49] |
| Overall | 2.23 [2.06, 2.41] | 2.31 [2.15, 2.47] | 2.16 [2.00, 2.31] | 2.18 [2.03, 2.34] | 2.22 [2.07, 2.37] |
| **Déjà vu** | | | | | |
| Critical lure | .222 [.095, .349] | .250 [.127, .373] | .111 [.032, .190] | .111 [.041, .181] | .174 [.088, .259] |
| Related lure | .075 [.027, .123] | .064 [.016, .111] | .061 [.021, .101] | .069 [.029, .110] | .067 [.028, .107] |
| Unrelated lure | .011 [.000, .022] | .025 [.005, .045] | .025 [.005, .045] | .006 [−.002, .013] | .017 [.004, .029] |
| Target | .076 [.019, .133] | .052 [.000, .104] | .072 [.021, .124] | .057 [.004, .111] | .064 [.013, .116] |
| Overall | .096 [.049, .143] | .098 [.052, .143] | .067 [.027, .108] | .061 [.026, .096] | .080 [.041, .120] |

conditions. Assumptions of sphericity were violated, $\chi^2(5) = 11.63, p = .040$, therefore degrees of freedom were corrected with Greenhouse-Geisser estimates of sphericity, using $\varepsilon = .819$. The effect of word condition on accuracy was significant, $F(2.46, 71.28) = 47.43$, $p < .001, \eta_p^2 = .621$, with accuracy lowest for critical lures, as anticipated (see Table 1 for descriptives).

To check whether our list manipulations influenced levels of erroneous familiarity, we next conducted a 2 x (novelty: high, low) x 2 (familiarity: high, low) repeated measures ANOVA on critical lure accuracy. There was a significant main effect of novelty, $F(1, 29) = 24.23, p < .001, \eta_p^2 = .455$, such that high novelty critical lures, $M = .714$ [.617, .811], were more accurately responded to than low novelty critical lures, $M = .414$ [.322, .506]. There was also a significant main effect of familiarity, $F(1, 29) = 51.11, p < .001, \eta_p^2 = .638$, such that high familiarity critical lures, $M = .475$ [.394, .556], were less accurately responded to than low familiarity critical lures, $M = .653$ [.583, .722]. Finally, there was a significant interaction between novelty and familiarity, $F(1, 29) = 11.45, p = .002, \eta_p^2 = .283$. Focusing on the main effects, it is evident that both list manipulations had the intended effects on levels of erroneous familiarity generated for critical lures—increased novelty salience decreased erroneous responding by drawing participants' attention to the objective novelty of the critical lure, whilst the lists selected from the *Stadler, Rodiger & McDermott (1999)*

**Peer**J _______________

norms for their elevated false alarm rates also demonstrated elevated false alarms in the current procedure.

### Déjà vu frequency

Déjà vu was reported at least once by 18 of the 30 participants (60%). Across all participants, the mean number of déjà vu reports was 12.83 [5.85, 19.82]. This value rose to 21.39 [11.41, 31.37] in the subsample who reported at least one déjà vu. In the following analyses, we analyse déjà vu frequency across the whole sample.

Déjà vu occurrence was assessed on a trial by trial basis. We were therefore able to calculate the likelihood with which a word from each condition would yield a déjà vu report (see Table 1 and Fig. 2A). Déjà vu frequency, as a proportion of all words presented within the given combinations of conditions, was assessed in a 2 (novelty) x 2 (familiarity) x 4 (word) repeated measures ANOVA. Assumptions of sphericity were violated for the main effect of word, $\chi^2(5) = 31.17, p < .001$, and the interactions between word x novelty, $\chi^2(5) = 92.60, p < .001$, word x familiarity, $\chi^2(5) = 54.54, p < .001$, and word x novelty x familiarity, $\chi^2(5) = 33.49, p < .001$. Degrees of freedom for these effects were corrected using Greenhouse-Geisser estimates of sphericity, using $\varepsilon = .564$, $\varepsilon = .413$, $\varepsilon = .482$ and $\varepsilon = .555$ respectively. Across all effects involving familiarity however, there were no significant differences, all $ps > .350$, suggesting that our manipulation of DRM strength did not influence déjà vu responses independently of the other factors. We therefore present the remaining effects involving the novelty and word conditions below.

There was a significant main effect of novelty, $F(1, 29) = 8.05, p = .008, \eta_p^2 = .217$, with a greater frequency of déjà vu reports under high novelty, $M = .097$ [.051, .142], than low novelty, $M = .064$ [.027, .101]. There was also a significant main effect of word, $F(1.69, 49.10) = 10.28, p < .001, \eta_p^2 = .262$, driven by the high frequency of déjà vu reports for critical lures (see Table 1 for descriptives).[1] Both of these findings are consistent with our hypotheses. Although there was no graded response according to familiarity condition, the lists contriving salient novelty generated the most déjà vu reports, likely driven by the greatest conflict between DRM-induced familiarity and novelty. Consistent with this interpretation, we were also able to show that déjà vu was reported more for critical lures than any other word condition.

The novelty x word interaction was also significant, and likely responsible for both main effects presented above, $F(1.24, 35.95) = 6.45, p = .011, \eta_p^2 = .182$. Figure 2 illustrates the homogeneity of responding within word conditions, which is broken only for critical lures. Crucially, critical lures in the low novelty condition remained comparable to other word conditions in their likelihood of yielding déjà vu reports, in the region of 10%, whereas those in the high novelty condition elicited déjà vu responding around 25% of the time. The presence of salient novelty, whereby participants were made aware that stimuli which they otherwise found to be familiar could not be so, appears important in elevating categorical reports of déjà vu within this procedure.

Our hypothesised elevation of déjà vu responses in maximal clash conditions was predicated upon participants correctly identifying critical lures as objectively new. To

[1] The high number of participants reporting no déjà vu experiences ensured that the déjà vu frequency data were highly positively skewed and likely do not satisfy parametric assumptions. We therefore present additional nonparametric tests of the main effects reported above. Wilcoxon's Signed Ranks tests found no effect of familiarity on déjà vu reports, $Z = -1.37, p = .171$, but there was a significant effect of novelty, $Z = -2.74, p = .006$. Friedman's test found a main effect of word, $\chi^2(3) = 17.04, p = .001$. Thus, the nonparametric equivalent tests of the main effects matched the patterns of significance obtained from parametric tests.

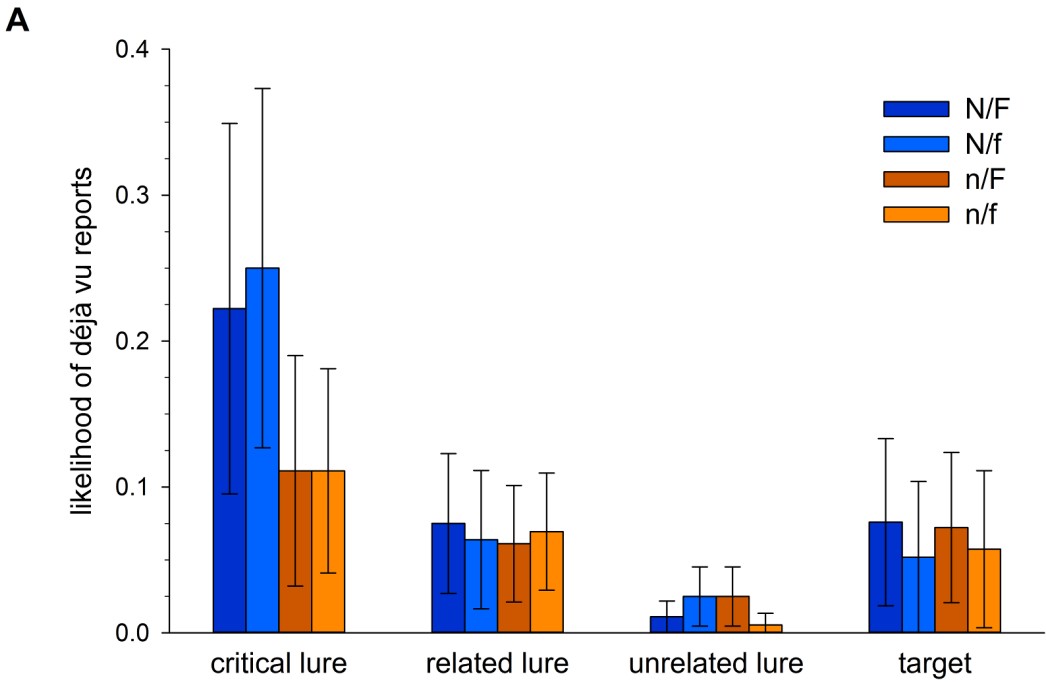

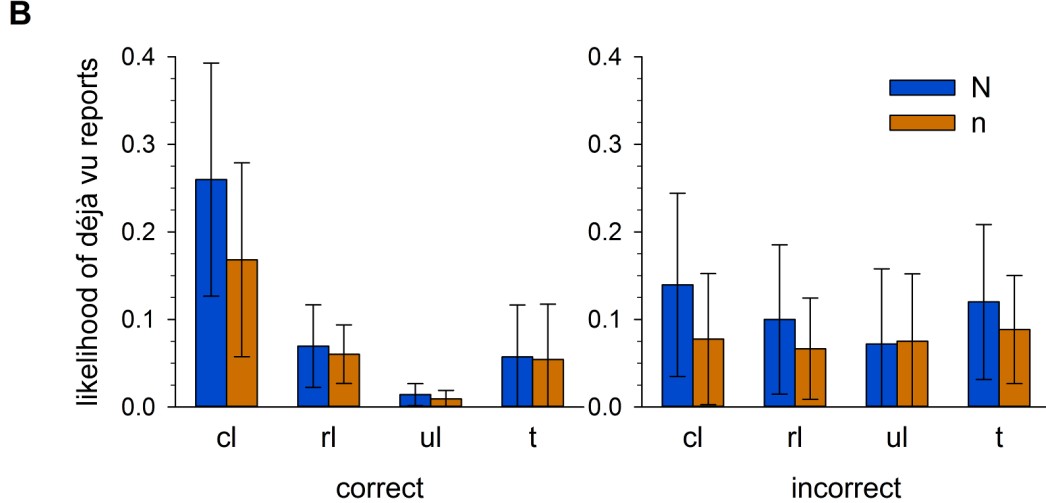

**Figure 2 Likelihood of déjà vu responding according to list and word condition.** (A) Shows the likelihood of a déjà vu response according to list and word type collapsed across recognition response correctness. Upper-case N indicates high novelty lists (blue hues), lower-case n indicates low novelty lists (orange hues). Upper-case F indicates high familiarity lists, lower-case f indicates low familiarity lists. (B) Shows the likelihood of a déjà vu response according to novelty manipulation (N is high novelty lists (blue), n is low novelty lists (orange)) and word type (cl, critical lure; rl, related lure; ul, unrelated lure; t, target), split according to correct (left panel) and incorrect (right panel) recognition responding. Error bars represent 95% CIs.

**Peer**J _______________

establish that the described pattern of déjà vu responses persisted in items to which participants had given correct recognition responses but not incorrect recognition responses, we recalculated déjà vu response likelihoods according to recognition response correctness (Fig. 2B). Given the ineffectiveness of the familiarity manipulation in influencing déjà vu frequency above, we collapsed across familiarity conditions so as to compare only high and low novelty lists by word condition. In a 2 (novelty) x 4 (word) repeated measures ANOVA on déjà vu responses to correctly identified words, assumptions of sphericity were violated for the main effect of word, $\chi^2(5) = 41.30$, $p < .001$, and the interaction, $\chi^2(5) = 105.37$, $p < .001$. Degrees of freedom for these effects were corrected using Greenhouse-Geisser estimates of sphericity, using $\varepsilon = .565$ and $\varepsilon = .385$ respectively. The main effect of novelty was no longer significant, $F(1, 26) = 3.63$, $p = .068$, $\eta_p^2 = .123$, though the main effect of word survived, $F(1.70, 44.11) = 8.85$, $p < .001$, $\eta_p^2 = .241$, again carried by the elevated déjà vu responding to critical lures.[2] The novelty x word interaction was no longer significant, $F(1.16, 30.06) = 2.16$, $p = .100$, $\eta_p^2 = .077$. In an equivalent ANOVA for incorrect responses there were no main effects of novelty, $F(1, 12) = 3.11$, $p = .103$, $\eta_p^2 = .206$, or word, $F < 1$, and no significant interaction, $F < 1$.[3] Although the previously described effects are attenuated when split according to recognition response correctness, there is nothing to suggest that the hypothesised elevation in déjà vu responses was driven by responses to critical lures which participants have incorrectly identified as old. There is therefore little indication that the overall findings relating to déjà vu frequency are driven by stimuli to which participants should be reporting no more than baseline levels of déjà vu.

## Déjà vu intensity

Déjà vu intensity was measured after the presentation of each word and coded as being rated from 1 (low) to 3 (high). We were interested in whether déjà vu, once reported, differed in intensity according to list condition. We again collapsed across familiarity conditions so as to compare déjà vu intensities across high and low novelty lists. We also restricted our analysis to critical lures, the stimuli in which déjà vu reports were most frequent, to avoid the problem of empty cells decimating the analysis. Fourteen participants reported déjà vu under both novelty conditions. A repeated measures $t$-test found no significant difference between déjà vu intensity ratings for high novelty critical lures, $M = 1.67$, $[1.31, 2.03]$, and low novelty critical lures, $M = 1.79$, $[1.38, 2.21]$, $t(13) = -0.69$, $p = .503$ $d = -.196$. (There were too few participants reporting déjà vu in both novelty conditions to warrant further analysis of the data subdivided according to correct [9 participants] and incorrect [4 participants] recognition responses.) Overall, déjà vu intensity did not vary according to novelty condition.

## Déjà vu descriptions

In order to quantitatively explore discursive accounts of naturalistic and experimentally-generated déjà vu experiences, we excluded participants who had never experienced déjà vu or who did not experience déjà vu in the experiment. This left us with a subsample of 15 participants. We conducted $n$-gram analyses on these participants' accounts of naturalistic

[2] Nonparametric tests of the main effects within correctly identified words found significant effects of both novelty, $Z = -2.74$, $p = .006$, and word, $\chi^2(3) = 12.77$, $p = .005$. Whilst the nonparametric test significances do not match their parametric equivalents above, they do match the parametric and nonparametric significances for the overall data, collapsed across correctness.

[3] Nonparametric tests within incorrectly identified words also found nonsignificant effects of novelty, $Z = -0.36$, $p = .723$, and word, $\chi^2(3) = 5.04$, $p = .169$.

and experimentally-generated déjà vu. Unigrams and bigrams which significantly differentiate naturalistic and experimental experiences of déjà vu are listed in Table 2, wherein positive $z$ values indicate $n$-grams which were used in the descriptions of previous déjà vu experiences significantly more than the experimental déjà vu experiences. Negative $z$ values indicate $n$-grams which showed the opposite pattern of correspondence.

A common thread across unigrams and bigrams was the presence of words describing the specificity of the déjà vu experience in question. Experimental reports of déjà vu were characterised by descriptions relating to the stimuli ("word[s]", "the word[s]"), modality of presentation ("seen", "had seen") and the setting ("experiment", "the experiment") e.g., "Sometimes, I had a very slight feeling that I had seen a particular word before". Naturalistic déjà vu experiences were more generalised ("situation", "experience") e.g., "Typically a scenario or situation I am in just seems familiar". In this respect, experimental reports of déjà vu appear to be in response to an experience which is more restricted to certain stimuli within the environment than naturalistic déjà vu experiences.

A related dimension along which there was the suggestion of differentiation was the duration of the experience. "Minutes" was used disproportionately to describe naturalistic experiences e.g., "A typical deja vu experience for me lasts a couple of minutes". "Seconds" appeared in the unthresholded table, but was not diagnostic of one or other category of experience (naturalistic: 4, experimental: 3, $z = .352$, $p = 1.00$) and was used in similar contexts across accounts e.g., naturalistic—"It lasted only a few seconds. . .," and experimental—"The deja vu last only a few seconds each time". In general, it would appear that the experimental experience was restricted to a shorter duration than that to which naturalistic déjà vu experiences can extend, though this generalisation did not fit with all participants' experiences e.g., the following from a description of the experimental déjà vu: "The duration was longer than previously experienced deja vu but also fainter".

More generally, these accounts offer insights into the nature of the experimentally-generated experience not afforded by the dichotomous or categorical response options available to them in the experiment itself. Specifically, some participants who used the toggle system to indicate that they had experienced déjà vu were much more cautious about describing the experimental experience as déjà vu when given the opportunity to explain their experience more precisely e.g., "I am unsure whether it was exactly deja vu but in some cases I saw words usually words I was expecting to see in the groups but did not, and it felt as if I had seen them". and "I am not really sure I had the two deja vu I reported, or if I think I had them only because it was the task of the experiment". Others reiterated what would be inferred from their categorical responses e.g., "The feeling of deja vu experience is quite strong but I only felt it when certain words came up, when it moved on to the next word… the feeling disappeared…". These accounts raise the question of whether demand characteristics and response acquiescence are important in influencing categorical responses in studies such as this one. Whilst this may not apply to all participants, there are clearly some participants for whom discursive response options give the experimenter a clearer idea of the inferences they should be making based on participant responses.
**Table 2** *n*-grams differentiating naturalistic and experimentally-generated déjà vu. The *n*-gram column shows unigrams and bigrams which (a) occurred with a frequency of at least the median (5 for unigrams, 4 for bigrams) across all text within the descriptions of naturalistic and experimental déjà vu occurrences and (b) were significantly disproportionately represented ($p < .05$) in one or other set of descriptions. The naturalistic and experimental headings quantify occurrences of the *n*-grams within the corresponding set of descriptions across all amalgamated accounts. *N* is the total count across both sets of descriptions, *z* is the binomial distribution *z* value calculated using the listed *n*-gram frequencies and an assumed binomial-*p* parameter of .5, and *p* is the probability of obtaining this *z* value by chance.

|  | *n*-gram | Previous | Experimental | *N* | *z* | *p* |
|---|---|---|---|---|---|---|
| **Unigrams** | have | 21 | 3 | 24 | 3.67 | 0.000 |
|  | usually | 12 | 1 | 13 | 3.05 | 0.001 |
|  | is | 18 | 4 | 22 | 2.98 | 0.001 |
|  | minutes | 6 | 0 | 6 | 2.45 | 0.007 |
|  | feel | 12 | 3 | 15 | 2.32 | 0.010 |
|  | any | 5 | 0 | 5 | 2.24 | 0.013 |
|  | situation | 5 | 0 | 5 | 2.24 | 0.013 |
|  | typical | 5 | 0 | 5 | 2.24 | 0.013 |
|  | experience | 13 | 4 | 17 | 2.18 | 0.015 |
|  | a | 34 | 19 | 53 | 2.06 | 0.020 |
|  | for | 8 | 2 | 10 | 1.90 | 0.029 |
|  | are | 6 | 1 | 7 | 1.89 | 0.029 |
|  | familiar | 6 | 1 | 7 | 1.89 | 0.029 |
|  | makes | 6 | 1 | 7 | 1.89 | 0.029 |
|  | same | 6 | 1 | 7 | 1.89 | 0.029 |
|  | before | 14 | 6 | 20 | 1.79 | 0.037 |
|  | me | 18 | 9 | 27 | 1.73 | 0.042 |
|  | but | 10 | 19 | 29 | −1.67 | 0.047 |
|  | whether | 3 | 9 | 12 | −1.73 | 0.042 |
|  | I | 64 | 87 | 151 | −1.87 | 0.031 |
|  | had | 11 | 24 | 35 | −2.20 | 0.014 |
|  | come | 0 | 5 | 5 | −2.24 | 0.013 |
|  | during | 0 | 5 | 5 | −2.24 | 0.013 |
|  | experiment | 0 | 5 | 5 | −2.24 | 0.013 |
|  | knew | 0 | 5 | 5 | −2.24 | 0.013 |
|  | new | 0 | 5 | 5 | −2.24 | 0.013 |
|  | old | 0 | 5 | 5 | −2.24 | 0.013 |
|  | did | 1 | 8 | 9 | −2.33 | 0.010 |
|  | sure | 0 | 6 | 6 | −2.45 | 0.007 |
|  | up | 0 | 8 | 8 | −2.83 | 0.002 |
|  | seen | 1 | 11 | 12 | −2.89 | 0.002 |
|  | was | 12 | 32 | 44 | −3.02 | 0.001 |
|  | were | 0 | 11 | 11 | −3.32 | 0.000 |
|  | the | 35 | 70 | 105 | −3.42 | 0.000 |
|  | word | 0 | 17 | 17 | −4.12 | 0.000 |
|  | words | 0 | 19 | 19 | −4.36 | 0.000 |

Table 2 (*continued*)

| | *n*-gram | Previous | Experimental | N | z | p |
|---|---|---|---|---|---|---|
| **Bigrams** | I have | 14 | 1 | 15 | 3.36 | 0.000 |
| | it is | 8 | 1 | 9 | 2.33 | 0.010 |
| | been in | 5 | 0 | 5 | 2.24 | 0.013 |
| | do not | 5 | 0 | 5 | 2.24 | 0.013 |
| | is not | 5 | 0 | 5 | 2.24 | 0.013 |
| | a typical | 4 | 0 | 4 | 2.00 | 0.023 |
| | it usually | 4 | 0 | 4 | 2.00 | 0.023 |
| | that my | 4 | 0 | 4 | 2.00 | 0.023 |
| | think that | 4 | 0 | 4 | 2.00 | 0.023 |
| | have been | 6 | 1 | 7 | 1.89 | 0.029 |
| | makes me | 6 | 1 | 7 | 1.89 | 0.029 |
| | come up | 0 | 4 | 4 | −2.00 | 0.023 |
| | I would | 0 | 4 | 4 | −2.00 | 0.023 |
| | the experiment | 0 | 4 | 4 | −2.00 | 0.023 |
| | the feeling | 0 | 4 | 4 | −2.00 | 0.023 |
| | the list | 0 | 4 | 4 | −2.00 | 0.023 |
| | what i | 0 | 4 | 4 | −2.00 | 0.023 |
| | word was | 0 | 4 | 4 | −2.00 | 0.023 |
| | words i | 0 | 4 | 4 | −2.00 | 0.023 |
| | words that | 0 | 4 | 4 | −2.00 | 0.023 |
| | during the | 0 | 5 | 5 | −2.24 | 0.013 |
| | I knew | 0 | 5 | 5 | −2.24 | 0.013 |
| | did not | 0 | 6 | 6 | −2.45 | 0.007 |
| | had not | 0 | 6 | 6 | −2.45 | 0.007 |
| | had seen | 0 | 6 | 6 | −2.45 | 0.007 |
| | it was | 1 | 9 | 10 | −2.53 | 0.006 |
| | the words | 0 | 7 | 7 | −2.65 | 0.004 |
| | in the | 1 | 10 | 11 | −2.71 | 0.003 |
| | I had | 5 | 20 | 25 | −3.00 | 0.001 |
| | the word | 0 | 13 | 13 | −3.61 | 0.000 |

## DISCUSSION

The modified DRM task reliably elicited categorical déjà vu reports. These déjà vu reports varied consistently, largely with our expectations—they were most likely to occur for inappropriately familiar words for which we contrived a clash between illusory familiarity and salient novelty. We showed an increased frequency of déjà vu reports when we elevated the awareness of objective novelty, but not when we elevated strength of DRM-induced familiarity alone. Examination of written accounts allowed us to contrast previously experienced naturalistic experiences of déjà vu with those resulting from the experimental procedures, suggesting that, whilst the experimentally-generated experience may approximate naturalistic déjà vu, it is more restricted in its focus. The written accounts also highlighted the potential influence of demand characteristics in elevating the frequency of déjà vu reports provided in categorical responses alone.

Our procedure captures a critical feature of déjà vu, the clash between subjective familiarity and objective novelty (*Brown, 2004*). Whilst our analyses compared the frequency of déjà vu reports according to the list-wise novelty manipulation, it should also be noted that each list, regardless of the novelty condition, included semantically related lures for which familiarity would have been high and novelty salience was low. Thus, we had both between-list and within-list controls for our high novelty critical lures (critical lures in the low novelty lists and related lures in the high novelty lists respectively). Both of these word conditions were highly familiar yet lacked verifiable novelty and tellingly, yielded significantly lower déjà vu reports than the critical stimuli. Stimuli in either one of these control conditions can be compared to those previously reported as generating déjà vu through familiarity without recollection alone (e.g., *Brown & Marsh, 2009*; *Cleary, Ryals & Nomi, 2009*) and the baseline levels of déjà vu reported for these stimuli, are consistent with the tendency for participants to report déjà vu under these circumstances. Importantly though, introducing verifiable novelty doubled déjà vu report rates, with this elevation seemingly driven by critical lures correctly identified as novel rather than those about whose objective status participants were confused. It remains to be seen whether contriving objectively verifiable novelty within the alternative procedures would elevate déjà vu responding further, as would be consistent with our operationalisation of the experience.

The precise role of objective novelty within the déjà vu experience remains to be established. It may be that novelty is absolutely necessary to establish that the familiarity in question is indeed illusory. Under these circumstances, the inappropriate familiarity signal should be indistinguishable from a conventional familiarity signal, with déjà vu emerging only from the combination of familiarity and novelty signals indicating that one of them must be wrong. Alternatively, verifiable novelty may help to confirm that the familiarity signal is illusory, though something carried in the familiarity signal alone—some intrinsic indicator that it is inappropriate—may be sufficient to achieve this. That déjà vu in the healthy population has no behavioural consequences and therefore that people tend always to discount the illusory familiarity signal (as opposed to déjà vécu) supports the second alternative. (At this point, it is worth noting that the association between elevated déjà vu responding and increased accuracy to critical lures in the current analogue mirrors the appropriate decision-making that accompanies naturalistic déjà vu experiences in the healthy population.) In any case, verifiable novelty tends to make for a very compelling argument that the familiarity experienced is inappropriate and may therefore lend itself to being told to others and remembered as an archetypal déjà vu experience. Déjà vu for an event which can never have happened before (e.g., Pasteur's funeral, *Berrios, 1995*) or in a country one has never previously visited (e.g., in France, *O'Connor, Lever & Moulin, 2010*) is bound to be more compelling than déjà vu during one's daily commute. Thus, the salient novelty with our experimental analogue may bring it closer to the déjà vu experiences people report to each other, than analogues without this component.

Despite the definitional improvement, the current procedure was still unable to generate déjà vu in 40% of participants, whilst eliciting a large number of déjà vu experiences

in the other 60%. It is unclear why the procedure is so variable in its effectiveness. One possibility, which draws on data from the discursive responses, is that some participants may view the experimentally-generated sensation as too restricted in its specificity to warrant being called déjà vu. The restriction to single word stimuli is a key component of the DRM procedure and it is difficult to see how this could be overcome whilst continuing to generate erroneous familiarity in this manner. Nonetheless, the use of secondary tasks to generate verifiable novelty could be successful in other procedures for which richer stimuli have been used to generate reports of déjà vu or erroneous familiarity (e.g., *Cleary, Ryals & Nomi, 2009*; *Brown & Marsh, 2008*). A comparison of experimental déjà vu report rates across stimuli of differing richness would offer insight into the degree to which stimulus complexity and coherence affects participants' willingness to report the experience, and therefore the face validity of the experimentally generated sensation itself.

The absence of a novelty effect on déjà vu intensity, is also worth considering here. There may be a number of reasons for this finding including: (i) that questions about déjà vu intensity are responded to more conservatively than questions about déjà vu occurrence; (ii) that once elicited, déjà vu intensity does not correspond to the strength of its trigger to elicit frequent déjà vu reports; and relatedly (iii) that déjà vu is an 'all-or-nothing' categorical process. That there were no floor effects in the compared intensity ratings argues against explanation (i). The other explanations however, warrant further investigation. Explanation (ii) could be falsified relatively easily. With refinement of the current procedure to generate déjà vu at varying frequencies, intensity ratings could be collected at each level, with a view to establishing a relationship between the two variables. The presence of a frequency-intensity relationship at some levels of frequency would suggest that the null finding here is caused by a failure to calibrate the current search appropriately. The absence of a relationship across all levels would suggest that the relationship between déjà vu intensity and the likelihood of déjà vu generation is not straightforward—a correspondence between the two continuous variables could either be absent or, according to explanation (iii), impossible. In this and previous work conducted in the lab, participants have been able to quantify the intensity of their déjà vu experiences on a continuous scale indicating that, counter to explanation (iii), déjà vu is not subjectively experienced as categorical. Similarly though, other memory experiences which are often conceptualised as categorical can be continuously quantified by participants (e.g., recollection; *Yonelinas, 1994*; *Mickes, Wais & Wixted, 2009*). Thus the nature of the déjà vu experience, as categorical or continuous in its presence and intensity, remains to be fully established.

Within this report, we have largely referred to the analysis of discursive responses as a counterpoint to the categorical self-reports of déjà vu collected during the modified DRM task. However, our *n*-gram analyses are also prone to bias from the question used to generate discursive accounts, which we may have introduced by asking participants about a "typical" past experience of déjà vu. Instead of detailing a specific episode which typifies their experience of déjà vu, many participants spoke in general terms about typical déjà vu experiences, which will undoubtedly have influenced our comparison of the two accounts.

We acknowledge that this method could be improved by simply modifying the question asked, but also suggest that further developments to this procedure may be valuable, especially in an attempt to better understand the subjective experience beyond simple extrapolation from categorical responses to questions about such unusual experiences. One such approach is to train support vector machines (SVMs) to more objectively classify discursive responses as belonging to one or other category of experience. This approach requires a larger corpus of text than was collected in this experiment, but which could be obtained if participants were asked to describe a greater number of previous and experimentally generated déjà vu experiences. *Selmeczy & Dobbins (2014)* successfully applied SVMs to demonstrate that linguistic content differs according to the recognition memory processes engaged at retrieval and we suggest that such approaches could also be applied to the study of déjà vu experiences.

Finally, despite progress towards a viable laboratory analogue, the pattern of déjà vu reports from the current experiments highlights a pervasive, problematic issue within the field. Whilst déjà vu was significantly more likely to be reported for critical lures, it was nonetheless also reported for other words. *O'Connor & Moulin (2010)* suggest that such non-hypothesised reporting (and therefore a proportion of hypothesised reporting) is driven by demand characteristics (*Orne, 1962*), a point highlighted by one participant in their discursive response. To minimise this artifact, O'Connor and Moulin suggested that déjà vu be assessed by post-experimental questionnaire alone, thereby reducing the trial-by-trial suggestion that déjà vu should be occurring. We found this impractical when seeking to identify specific words triggers of déjà vu reports, and implemented a toggle system using which participants could ignore the question of déjà vu occurrence until it became pertinent. Nonetheless, the persistent cue may still have acted to reinforce acquiescent responding. Alternative methods of questioning which afford both trial-level specificity and minimal pressure to acquiesce would add further credibility to reports proposing laboratory analogues of déjà vu. In their absence however, reporting procedures which allow participants to contextualise their responses go some way towards clarifying the features of a naturalistic experience that are both well and poorly represented by analogues such as this.

## ACKNOWLEDGEMENTS

We are thankful to Ronan Kearney for his work prompting development of the déjà vu toggle procedure.

### Funding

Akira O'Connor is supported by a SINAPSE (Scottish Imaging Network: A Platform for Scientific Excellence) fellowship. Josephine Urquhart was supported by the University of St Andrews URIP Scheme. Funds for participant compensation were provided by the School of Psychology and Neuroscience, University of St Andrews. The funders had no

role in study design, data collection and analysis, decision to publish, or preparation of the manuscript.

## Grant Disclosures

The following grant information was disclosed by the authors:
Scottish Imaging Network: A Platform for Scientific Excellence.
University of St Andrews URIP Scheme.
School of Psychology and Neuroscience, University of St Andrews.

## Competing Interests

The authors declare there are no competing interests.

## Author Contributions

- Josephine A. Urquhart conceived and designed the experiments, performed the experiments, analyzed the data, wrote the paper, reviewed drafts of the paper.
- Akira R. O'Connor conceived and designed the experiments, analyzed the data, wrote the paper, prepared figures and/or tables, reviewed drafts of the paper.

## Human Ethics

The following information was supplied relating to ethical approvals (i.e., approving body and any reference numbers):

University Teaching and Research Ethics Committee at the University of St Andrews: Approval Number PS10697.

## Supplemental Information

Supplemental information for this article can be found online at http://dx.doi.org/10.7717/peerj.666#supplemental-information.

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
