# Peer review of "The awareness of novelty for strangely familiar words: a laboratory analogue of the déjà vu experience"

_PeerJ, doi:10.7717/peerj.666_

## Round 0.1 · original submission · Minor Revisions

Like the two reviewers, I found this an intriguing and novel study. I particularly appreciated the nuanced discussion and the fact you haven't oversold your results. However, like the reviewers, I found a few points that were less than perfectly clear. I also had as a couple of more substantive concerns of my own.

I'd like to see the following issues addressed in the revised manuscript (or in your rebuttal if that is more appropriate), alongside the issues raised by the reviewers. Please pay particular attention to Reviewer 1's helpful suggestions re clarifying the methods:

1. Deja vu requires that the subject experiences familiarity for a word they know to be objectively novel. That is, participants should only indicate that deja vu was experienced for items that they also rejected as being novel. However, unless I've missed something, the analyses consider all reports of deja vu, irrespective of whether the item was reported as "new". How often did participants (erroneously) indicate experiencing "deja vu" when in fact they had already indicated that the item was "old"? Does the higher incidence of "deja vu" reports for critical lures and for high novelty lists survive if these erroneous deja vu reports are excluded? Answering these questions would address Reviewer 1's concern that there was no independent measure of novelty.

2. The number of deja vu reports is relatively low, with 40% of participants giving no reports of deja vu at all. This means that the data are unlikely to be even close to normally distributed and, therefore, that ANOVA is inappropriate. Obviously, re-analysing the data using non-parametric statistics would prevent consideration of interactions. However, it would be helpful to report non-parametric analyses of the significant effects that come out of the ANOVA (i.e., deja vu responses are more frequent for critical lures than for other words; and for high- than low-novelty lists).

NB An alternative approach to explore for future studies might be to use logistic regression with mixed random effects, but I wouldn't expect you to re-analyse the data for the current submission.

Typo: The text refers to Figure 2 when in fact there is only one figure.

Reviewer 1 ·

Basic reporting

No comments

Experimental design

No comments

Validity of the findings

The authors assert throughout this manuscript that their laboratory technique for recreating deja vu produces objective awareness that certain words are novel, along with a subjective feeling of familiarity. However, there is no independent measure of whether subjects were aware that certain words were novel. During the test phase, subjects are reminded of the character string and their response – does this remain on the screen throughout the entire test phase? Subjects might pay no attention to it, focusing instead on their task of deciding old vs. new as well as the déjà vu status bar. Is there any way to index how successful the novelty manipulation was at creating objective knowledge and awareness that a word was novel? It seems possible that subjects were not aware that some of the words were novel. Could high novelty words simply be having an implicit impact on responses (i.e., be occurring outside of awareness)? This points to the need for some independent way of assessing whether subjects were consciously aware that some of the words had not been presented before. If they were unaware, it might explain their reluctance to describe their experience as déjà vu. I feel that this is an important issue that should be addressed, possibly in the discussion.

Additional comments

This is an interesting paper about an unusual and fascinating experience. It has been conducted rigorously and is well-written. I have a few minor comments for the authors to consider.

Introduction

The descriptions of previous attempts to generate a lab model are very brief and involve a lot of jargon, e.g. “subthreshold stimulus presentation” and “butcher on the bus.” Some elaboration and clarification would help.

Similarly, the brief description of the current study at the end of the intro is unclear. For instance, at this stage I did not know what was meant by an “unstudied critical lure”, “illusory recognition”, “critical lure novelty”, nor what was meant by varying “familiarity orthogonally at list level.” This needs to be explained in much simpler terms before getting into technical terminology. I was hoping to find a clear explanation of how subjects could be objectively aware that an item was novel despite having a feeling of familiarity.

Method

Stimuli and materials section assumes that the reader knows the basics of the DRM paradigm.

Could be clearer by informing the reader that there were 24 study lists and 24 test lists.

In the pre-experimental questionnaire asking about naturalistic déjà vu experiences, subjects are asked to comment on a “typical” déjà vu experience, whereas postexperimentally, they were asked to comment on a specific déjà vu experience (ie. They task they had just completed). Why weren’t subjects asked about a specific instance in the pre-experimental questionnaire? Is déjà vu so fleeting and vague that specific instances in the last year would be too difficult to remember? Some acknowledgement of this difference should be included in the paper.

At the start of the design and procedure section, it would be clearer to know how high vs. low novelty and familiarity were operationalized, before explaining the within and between subjects factors.

Could examples of word lists for all 4 familiarity and novelty combinations be included in an Appendix?

Might be nice to explain what an n-gram is! Likewise for bigrams and unigrams!

Results

Table 2 is large and not particularly informative. I would prefer to simply read about the n-gram analyses in the text, along with relevant examples.

Discussion

It might be worth considering the impact of asking subjects about previous, naturalistic déjà vu experiences before the experiment. This primes subjects and places strong social demands upon them since they would then know that the experimenters were attempting to recreate déjà vu.

·

Basic reporting

No comment.

Experimental design

No comment.

Validity of the findings

12 of the participants did not report DV. 18 did. However among these 18, the mean number of DV was 21.6 which is puzzling. Why would some subject be quite prone to report DV while the others would not at all? This point merits clarification.

Additional comments

I really enjoyed reading this study. It is very clearly written and the rationale for it and the experiment are clever. The main question of course is how the DV elicited by the experiment relates to the real-life DV. The authors are quite cautious about this relationship. They draw their conclusion on this point mainly from the result of their n-gram analysis. I must say however that I have found this part to be quite weak since readers are left to read and then "interpret" the results (Table 2). I appreciate the attempt the authors have made with this new analysis but think they should propose, for the future, new (more efficient?) ways to study this relationship, which is at the core of the field
.
In the same vein, some authors have discussed whether DV could apply to "single stimuli" such as words or pictures whereas DV usually concerns real-life whole situations. It could also be worthwhile to discuss this and propose ways to make progress on experiments that would be closer to this real-life situation.

---

## Round 0.2 · accepted · Accept

All of the issues the reviewers and I raised have been comprehensively addressed. Thank you for making my job easy!

Reviewer 1 ·

Basic reporting

No comments

Experimental design

No comments

Validity of the findings

No comments

Additional comments

I am happy with the revisions the authors have made to the manuscript